# Leaf Morpho-Colorimetric Characterization of Different Grapevine Varieties through Changes on Plant Water Status

Gastón Gutiérrez-Gamboa [1], Nicolás Torres-Huerta [2], Miguel Araya-Alman [3,*], Nicolás Verdugo-Vásquez [4], Yerko Moreno-Simunovic [5], Héctor Valdés-Gómez [6] and César Acevedo-Opazo [5,*]

1. Escuela de Agronomía, Facultad de Ciencias, Universidad Mayor, Huechuraba 8580000, Chile; gaston.gutierrez@umayor.cl
2. Stoller Chile SA, Ruta 5 Sur, KM 192, Curicó 3340000, Chile; nicolastorreshuerta@hotmail.com
3. Departamento de Ciencias Agrarias, Universidad Católica del Maule, km 6 Camino Los Niches, Curicó 3340000, Chile
4. Centro de Investigación Intihuasi, Instituto de Investigaciones Agropecuarias (INIA), Colina San Joaquín s/n, La Serena 1700000, Chile; nicolas.verdugo@inia.cl
5. Facultad de Ciencias Agrarias, Universidad de Talca, 2 Norte 685, Casilla 747, Talca 346000, Chile; ymoreno@gmail.com
6. Departamento de Fruticultura y Enología, Pontificia Universidad Católica de Chile, Av. Vicuña Mackenna 4860, Macul 7810000, Chile; hevaldes@uc.cl
* Correspondence: marayaa@ucm.cl (M.A.-A.); cacevedo@utalca.cl (C.A.-O.); Tel.: +56-9-5074-5436 (M.A.-A.); +56-9-9849-0066 (C.A.-O.)

**Abstract:** (1) Background: Currently, some ampelographic methods are developing in order to identify grapevine varieties. For this purpose, morpho-colorimetric parameters in leaves have been analyzed by digital imagen analysis, but some environmental conditions may affect their determinations. (2) Methods: A research study was conducted to characterize leaf morpho-colorimetric parameters in five grapevine varieties growing under different plant water status and to discriminate them under these conditions. Leaves were collected in vines, and twelve leaf morpho-colorimetric and fractal dimension variables were assessed. (3) Results: Merlot presented the highest values of perimeter and area to perimeter ratio in leaves and higher leaf area than Chardonnay in both plant water conditions. Most of the leaf morpho-colorimetric variables allowed discriminating the grapevine varieties under the contrasted hydric conditions. Under non-water stress, Carmenère was not related to any measured parameters. Merlot was positively related to most of the leaf morphometric parameters, whereas Chardonnay presented the opposite behavior. RGB color system variables allowed discriminating the grapevine varieties under water stress conditions, and Sauvignon Blanc was not related to any measured parameter. Chardonnay and Pinot Noir were positively related to green color and negatively related to most of the leaf morphometric parameters, whereas Merlot showed the opposite behavior. (4) Conclusions: Leaf morpho-colorimetric and fractal dimension parameters were affected by plant water stress and more variables should be incorporated into the new ampelographic methods in order to characterize leaf morpho-colorimetric parameters of the different grapevine varieties more clearly.

**Keywords:** ampelography; eccentricity; fractal dimension; morphometric characteristics; RGB color system; water stress

## 1. Introduction

Ampelography is the field of botany concerning the identification and classification of grapevine varieties that includes several parameters or descriptors that are measured on leaves, shoots, clusters and berries [1,2]. The main descriptors involved in ampelographic characterization comprises types of hair, color, shape and size, textures and appearance [3]. Based on this, ampelography provides relevant morphological and

agronomical information for varietal characterization studies, breeding programs and conservation purposes [4,5].

Ampelography remains a problem of concern since the wide number of grapevine varieties and the variability within them cause difficulty in their identification and recognition [6]. More than 6000 varieties have been identified based on ampelographic descriptors, but only a small number of varieties are used for the vineyard establishment [7]. Currently, there is great interest within the wine industry to explore the potential of minority grapevine varieties as adaptation strategies to face global warming [8]. Based on this, there is a need to ensure trueness of plant and to avoid planting wrong material, which may result in significant financial losses to viticulturists.

A more objective classification of grapevine varieties has been developed in the last decades by chemical, spectroradiometry and genetic fingerprinting techniques, including recently digital photography methods [2,6,7]. Based on empirical knowledge, leaf shape and shoot-tip pubescence are the best ampelographic descriptors for variety identification [1,9]. Therefore, leaf morphology provides distinctive traits for reaching this goal [1,9]. In recent years, some authors reported that plant water stress may affect leaf morphological characteristics, affecting varietal identification [2,10]. By digital photography and image analysis algorithms, it is possible to obtain a range of leaf morphological parameters that may solve the above-mentioned problems [11]. In a preliminary study, Fuentes et al. [6] reported that the automatic extraction of morpho-colorimetric data, NIR chemical fingerprinting and machine learning modelling rendered rapid, accurate and non-destructive methods for variety classification. However, it is possible that leaf morpho-colorimetric characteristics of the vines could be affected by plant water status. Therefore, the aim of this work was to characterize morpho-colorimetric and fractal dimension parameters from scanned mature leaves of vines managed under different soil water status by image analysis and to discriminate them under these contrasting conditions.

## 2. Materials and Methods

### 2.1. Site of Study and Plant Material

An experimental vineyard (cv. Sauvignon Blanc, Chardonnay, Carmenère, Merlot and Pinot Noir) was used for this field trial during the 2011–2012 growing season. The experimental vineyard is in Panguilemo, Talca, Chile (Maule Valley) (Talca, WGS84 datum, 35°22.2′ S; 71°35.39′ W, at 121 m.a.s.l.). The vineyard was established in 2006, and the vines were trained to a vertical shoot position trellis system and pruned into two bilateral spur cordons. The vine density was about 5000 vines ha$^{-1}$, spaced at 2.00 m × 1.00 m between rows and within the row, respectively, with an east–west orientation. The vineyard was drip irrigated using two lines of 2 L h$^{-1}$ with self-compensating emitters spaced every 0.5 m.

### 2.2. Soil and Climate Conditions

The vineyard soil presents a clay loam texture with a depth root of 1.50 m. Soil bulk density, field capacity, wilting point and available water were 1.36 g cm$^{-3}$, 0.31 m$^3$ m$^{-3}$, 0.13 m$^3$ m$^{-3}$ and 0.18 m$^3$ m$^{-3}$, respectively. The reports published by Pañitrur-De la Fuente et al. [12] and Gutierrez-Gamboa et al. [13] present more information concerning soil data and viticultural management performed in the vineyard.

An automatic weather station (Adcon Telemetry, A730, Klosterneuburg, Austria) located close to the experimental vineyard (around 50 m from the field trial) was used to obtain climatic data [13]. Briefly, average, maximum and minimum temperature in the growth season were 16.9, 34.6 °C and 0.0 °C, respectively, whereas precipitations registered for the same period was 31.8 mm, which was concentrated in the spring months. In the winter, 1438 chilling hours were accumulated. In the growth season, degree days accumulation was 1375 °C, vapor-pressure deficit reached 1.05 kPa and the reference evapotranspiration was 1037 mm. The climatic characteristics mentioned are characteristic of the Maule valley [14].

### 2.3. Experimental Design

The experimental design was a randomized complete block divided into two treatments (blocks) in which five grapevine varieties, such as Sauvignon Blanc, Chardonnay, Carmenère, Merlot and Pinot Noir, were arranged. One of the two blocks was managed under no water stress (without irrigation restriction), while the other was managed following a progressive water restriction until reaching severe water stress (leaf water potential $< -1.4$ MPa), according to the stress classification stated by van Leeuwen et al. [15]. A total of ten treatments (variety $\times$ water condition) were arranged in the vineyard, considering five plants per treatment in which eight mature leaves per plant were collected to determine leaf morpho-colorimetric variables. The selected vines evidenced good phytosanitary conditions and were homogeneous in vegetative growth, productivity and climatic conditions.

### 2.4. Leaf Collection Process and Leaf Morpho-Colorimetric and Fractal Dimension Analysis

Leaf collection protocol was carried out following the methodology stated by Fuentes et al. [6]. Briefly, eight mature leaf samples by plant were collected from the vineyard, stored in plastic bags and then transported in a cooler with ice blocks to avoid leaf dehydration. Healthy, fully expanded and mature leaves (including petiole) were collected from the fifth position of each vine shoot with the aim of attaining uniform physiological maturity of leaves for modelling purposes.

The collected leaves were scanned using a Hewlett Packard Scanjet G3010 (Hewlett-Packard Software Company, Palo Alto, CA, USA) scanner (Figure 1).

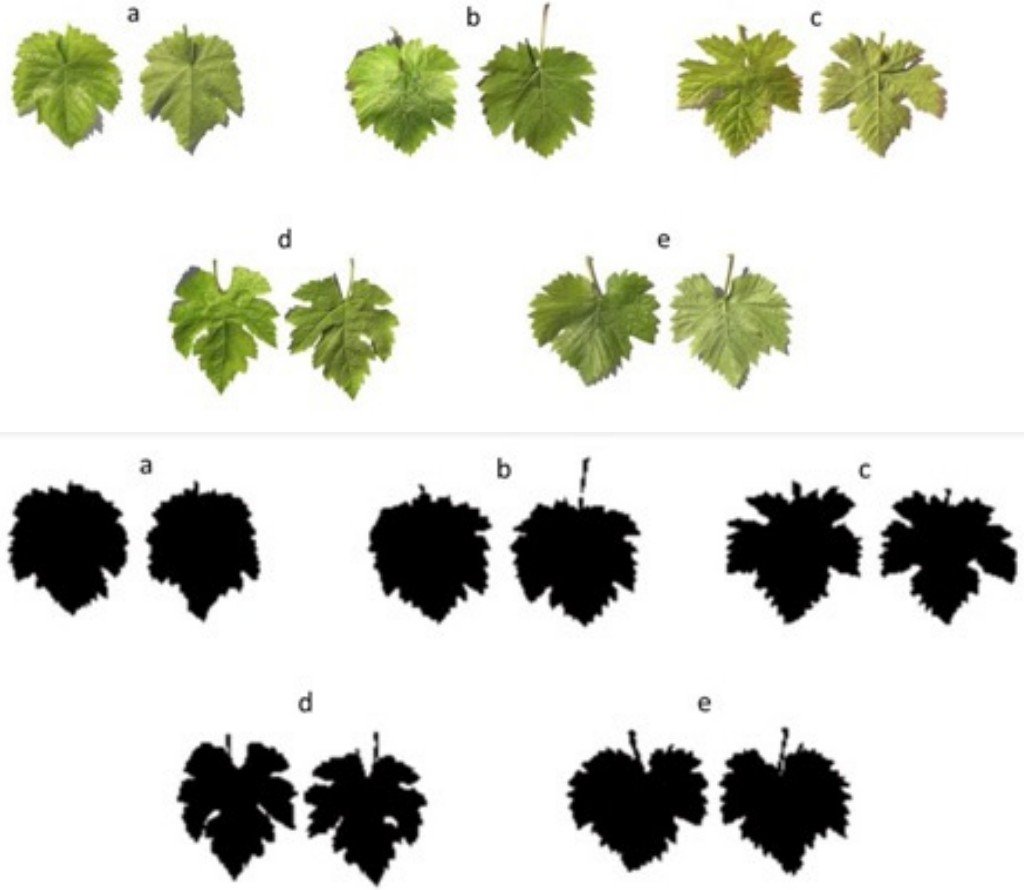

**Figure 1.** Example of scanned leaves of the 5 cultivars for the classification study (**up**) and corresponding binary images for automatic shape recognition (**down**) to obtain morpho-colorimetric parameters from each leaf. Letter of each leaf corresponds to (**a**) Sauvignon Blanc, (**b**) Chardonnay, (**c**) Carmenère, (**d**) Merlot and (**e**) Pinot Noir. Leaf sizes in the Figure are not proportional to real values.

The obtained images were analyzed by using a customized code written in Matlab® ver. R2010a (Mathworks Inc., Natick, MA, USA). The blue band of the image was then post-processed to identify the leaf morpho-colorimetric and fractal dimension variables (Figure 1). As described by Fuentes et al. [6], an initial calibration for the scanner was made using black squares as references of known dimensions to relate pixel count in the x and y coordinates to dimensions and area using metric units (cm and cm$^2$). Scanner calibration allowed automatic analysis of leaf images, extracting morphometric and color parameters and the fractal dimensions of each leaf sample [6]. These variables were analyzed according to the protocols stated by Fuentes et al. [6]. The parameters obtained from this process are presented in Table 1.

**Table 1.** Description of the leaf morpho-colorimetric features for variety characterization and discrimination.

| Parameter | Unit | Description |
|---|---|---|
| Area (A) | cm$^2$ | Area of the leaf surface |
| Perimeter (L) | cm | Perimeter of the leaf surface |
| Maximum length (LMax) | cm | Maximum length of the leaf |
| Minimum length (LMin) | cm | Minimum length of the leaf |
| Maximum length of petiole (LMax P) | cm | Length of leaf vine petiole |
| Area to perimeter ratio (P$^2$/A) | Unitless | Area to perimeter ratio of leaf |
| Eccentricity (Ex) | Unitless | Length: width ratio in which zero is given to a straight line and one represents a circle |
| RGB color scale | Unitless | RGB scale used to measure color: R = red, G = green, B = blue colors and RG = red and green index |
| Fractal dimension (FR) | Unitless | Fractal dimension measured using the box-counting method |

### 2.5. Plant Water Status and Gas Exchange Measurements

The leaf water potential (Ψleaf) was measured to define the treatments. This parameter was determined by using a pressure chamber (PMS Instrument Co., model 600, Corvallis, OR, USA), according to the methods stated by Acevedo-Opazo et al. [16] and by Jara-Rojas et al. [17].

An infrared gas analyzer model LI 6400 (Li-cor, Lincoln, NV, USA) was used to determine stomatal conductance ($g_s$), transpiration (E) and net $CO_2$ assimilation ($A_N$) in Sauvignon Blanc, Chardonnay, Merlot and Pinot Noir and not for Carmenère. Based on some of these variables, the intrinsic water use efficiency ($A_N g_s^{-1}$) was calculated as the ratio between $A_N$ and $g_s$ according to the report published by Medrano et al. [18]. The gas exchange measurements were performed according to the steps reported by Gutiérrez-Gamboa et al. [13]. Three determinations were made between 12:00 and 14:00 h in the north side of the canopy at the middle zone (sixth leaf from the tip on fruit-bearing shoots) on 20 December 2011, 27 December 2011 and 21 January 2012. Measurements were performed in five different vines per treatment on fully sunny days without changing the original position of the selected leaves in the canopy and maintaining the same light regime, ensuring that the leaves were exposed to full sunlight (PAR > 800 μmol m$^{-2}$ s$^{-1}$).

### 2.6. Statistical Analysis

The measured variables were analyzed by principal component analysis (PCA) to classify the leaves by variety using The Unscrambler® X ver. 10.1 (CAMO Software, Oslo, Norway). PCA was performed to identify the main relationships among the measured variables and to relate them to the quality of characterization (prediction) of the different grapevine varieties. Following this assumption, the proposed method considers that the first and second principal components of the PCA (PC 1 and PC 2) should represent the leaf morpho-colorimetric characterization at each observation (variety) if a significant amount of variance is explained by PC1 and PC2. If so, the score of both components can be used to rank all the observations according to the leaf morpho-colorimetric parameter values observed at these sites. In addition, the significant differences for the measured parameters

were assessed using analysis of variance (ANOVA), and the Tukey test ($p \leq 0.05$) was used for mean separation using the Statgraphics Centurion XVI.I statistical package (Warrento, VA, USA). Finally, the contribution of each variable to each principal component was added as Supplementary Materials.

## 3. Results

### 3.1. Leaf Morpho-Colorimetric Variables Analyzed in Grapevine Varieties Cultivated under No Water Stress Conditions

Leaf morpho-colorimetric parameters such as eccentricity, blue color, red and green index and fractal dimension were statistically similar among the varieties (Table 2). Merlot presented the highest values of leaf perimeter, maximum length of leaf and area to perimeter ratio (Table 2). In contrast, Pinot Noir exhibited the lowest value of this last parameter (Table 2). Merlot showed higher leaf area than Chardonnay and Sauvignon Blanc and higher red and green index than Carmenère and Chardonnay (Table 2). Maximum length of petiole was higher in Carmenère, Merlot and Sauvignon Blanc than in Chardonnay (Table 2). Regarding color parameters, Carmenère and Sauvignon Blanc presented higher red and green color than Merlot and Pinot Noir (Table 2).

**Table 2.** Mean values for the leaf morpho-colorimetric features obtained by scanning leaves from five different grapevine varieties under stress and no stress treatments.

| Parameter | Treatment | Variety | | | | |
|---|---|---|---|---|---|---|
| | | Carmenère | Chardonnay | Merlot | Pinot Noir | Sauvignon Blanc |
| Area (cm$^2$) | No Stress | 161.37 ± 18.55 ab | 137.08 ± 6.58 a | 196.05 ± 14.85 b | 160.87 ± 5.71 ab | 152.22 ± 15.67 a |
| | Stress | 143.92 ± 10.14 a | 129.74 ± 2.38 a | 180.87 ± 18.62 b | 121.15 ± 18.29 a | 147.21 ± 0.68 ab |
| Perimeter (cm) | No Stress | 83.01 ± 6.10 a | 75.77 ± 2.30 a | 98.17 ± 4.23 b | 74.57 ± 1.78 a | 77.28 ± 3.70 a |
| | Stress | 80.29 ± 3.11 a | 74.67 ± 1.09 a | 97.80 ± 9.54 b | 67.12 ± 5.55 a | 79.65 ± 3.22 a |
| Maximum length (cm) | No Stress | 13.10 ± 0.74 a | 12.36 ± 0.24 a | 14.62 ± 0.46 b | 13.20 ± 0.28 a | 12.73 ± 0.71 a |
| | Stress | 12.43 ± 0.44 a | 12.09 ± 0.06 a | 14.09 ± 0.71 b | 11.53 ± 0.91 a | 12.64 ± 0.42 ab |
| Minimum length (cm) | No Stress | 12.30 ± 0.76 a | 11.31 ± 0.32 a | 13.76 ± 0.60 b | 12.24 ± 0.24 a | 11.86 ± 0.57 a |
| | Stress | 11.62 ± 0.38 a | 11.01 ± 0.13 a | 13.08 ± 0.68 b | 10.55 ± 0.79 a | 11.64 ± 0.37 a |
| Maximum length of petiole (cm) | No Stress | 11.78 ± 1.01 b | 9.00 ± 0.29 a | 11.80 ± 0.85 b | 10.12 ± 0.66 ab | 11.40 ± 1.25 b |
| | Stress | 10.76 ± 0.43 b | 9.01 ± 0.23 ab | 11.05 ± 0.64 b | 7.93 ± 1.18 a | 10.45 ± 0.94 b |
| Area to perimeter ratio | No Stress | 42.76 ± 1.40 b | 41.89 ± 0.67 b | 49.23 ± 2.44 c | 34.57 ± 0.66 a | 39.32 ± 0.50 b |
| | Stress | 44.88 ± 2.59 a | 42.98 ± 0.54 a | 52.90 ± 4.98 b | 37.33 ± 0.51 a | 43.19 ± 3.07 a |
| Eccentricity | No Stress | 0.35 ± 0.01 a | 0.41 ± 0.00 a | 0.35 ± 0.04 a | 0.39 ± 0.03 a | 0.37 ± 0.03 a |
| | Stress | 0.35 ± 0.01 a | 0.42 ± 0.02 b | 0.38 ± 0.02 ab | 0.41 ± 0.03 b | 0.40 ± 0.00 ab |
| Red color | No Stress | 69.48 ± 2.28 c | 68.96 ± 0.63 bc | 63.58 ± 2.35 a | 64.37 ± 0.95 ab | 71.83 ± 1.80 c |
| | Stress | 74.93 ± 2.50 b | 81.24 ± 2.49 b | 68.16 ± 1.09 a | 75.92 ± 3.47 b | 75.94 ± 2.44 b |
| Green color | No Stress | 75.06 ± 3.05 c | 74.35 ± 1.73 bc | 65.39 ± 3.39 a | 67.72 ± 1.82 ab | 75.74 ± 1.96 c |
| | Stress | 81.48 ± 2.74 bc | 85.36 ± 2.55 c | 70.91 ± 0.71 a | 78.97 ± 3.58 bc | 77.77 ± 1.72 b |
| Blue color | No Stress | 46.54 ± 0.76 a | 46.86 ± 1.69 a | 48.53 ± 0.62 a | 47.53 ± 1.52 a | 49.06 ± 0.49 a |
| | Stress | 45.78 ± 0.63 a | 50.34 ± 0.83 b | 47.81 ± 1.52 ab | 48.04 ± 1.40 ab | 49.49 ± 1.10 b |
| Fractal dimension | No Stress | 1.65 ± 0.02 a | 1.52 ± 0.02 a | 1.64 ± 0.04 a | 1.65 ± 0.02 a | 1.60 ± 0.03 a |
| | Stress | 1.67 ± 0.02 a | 1.67 ± 0.01 a | 1.63 ± 0.03 a | 1.49 ± 0.03 a | 1.64 ± 0.01 a |
| Red and green index | No Stress | −0.039 ± 0.005 a | −0.037 ± 0.007 a | −0.014 ± 0.009 b | −0.025 ± 0.009 ab | −0.027 ± 0.005 ab |
| | Stress | −0.042 ± 0.010 a | −0.025 ± 0.004 ab | −0.020 ± 0.010 b | −0.020 ± 0.006 b | −0.012 ± 0.005 b |

For each variable and treatment, different letters in the same row show statistically significant differences assessed using the Tukey test ($\alpha = 0.05$).

### 3.2. Leaf Morpho-Colorimetric Variables Analyzed in Grapevine Varieties Cultivated under Water Stress Conditions

Fractal dimensions did not vary among the varieties cultivated under water stress and no water stress conditions (Table 2). Merlot presented the highest leaf perimeter, minimum length of leaf and area to perimeter ratio and the lowest red color (Table 2). Merlot showed higher leaf area and maximum length of leaf than Carmenère, Chardonnay and Pinot Noir (Table 2). Carmenère presented lower eccentricity than Chardonnay and Pinot Noir, while this last one showed lower maximum length of petiole than Carmenère, Merlot and Sauvignon Blanc (Table 2). Regarding color parameters, Chardonnay presented higher green color than Merlot and Sauvignon Blanc (Table 2). Carmenère showed lower blue color than Chardonnay and Pinot Noir and lower red and green index than Merlot, Pinot Noir and Sauvignon Blanc (Table 2).

### 3.3. Leaf Gas Exchange Parameters and Intrinsic Water Use Efficiency in Grapevine Varieties Managed under Different Water Supply

Leaf gas exchange parameters were not affected in grapevine varieties managed under stress water conditions (Table 3). Only the grapevine varieties cultivated under no water stress conditions presented statistical differences on transpiration (E), stomatal conductance ($g_s$) and intrinsic water-use efficiency ($A_N$ $g_s^{-1}$) (Table 3). Based on this, Pinot Noir presented lower transpiration than Merlot and Sauvignon Blanc (Table 3). Sauvignon Blanc showed higher stomatal conductance and lower intrinsic water-use efficiency than Merlot and Pinot Noir (Table 3).

**Table 3.** Gas exchange variables and intrinsic water-use efficiency obtained in leaves from five different grapevine varieties under stress and no stress treatments.

| Parameter | Treatment | Variety | | | |
|---|---|---|---|---|---|
| | | Chardonnay | Merlot | Pinot Noir | Sauvignon Blanc |
| Net $CO_2$ assimilation ($A_N$) | No stress | 16.74 a | 16.36 a | 16.39 a | 17.39 a |
| | Stress | 10.90 a | 11.04 a | 10.33 a | 13.30 a |
| | Both | 13.82 a | 13.70 a | 13.36 a | 15.35 a |
| Transpiration (E) | No stress | 11.01 ab | 11.81 b | 9.47 a | 12.23 b |
| | Stress | 7.49 a | 7.37 a | 6.45 a | 7.79 a |
| | Both | 9.25 a | 9.59 a | 7.97 a | 10.00 a |
| Stomatal conductance ($g_s$) | No stress | 0.42 ab | 0.40 a | 0.38 a | 0.54 b |
| | Stress | 0.25 a | 0.22 a | 0.20 a | 0.30 a |
| | Both | 0.33 a | 0.31 a | 0.34 a | 0.42 a |
| Intrinsic water-use efficiency ($A_N$ $g_s^{-1}$) | No stress | 40.30 ab | 41.24 b | 48.06 b | 32.40 a |
| | Stress | 68.02 a | 70.95 a | 65.69 a | 67.89 a |
| | Both | 50.14 a | 56.09 a | 56.87 a | 50.14 a |

For each variable and treatment, different letters in the same row show statistically significant differences assessed using the Tukey test ($\alpha = 0.05$). Data correspond to the average of the three data of measurements.

### 3.4. Multivariate Analysis to Discriminate the Different Grapevine Varieties

In order to discriminate the different varieties and assess the effects of treatments on leaf a morpho-colorimetric characteristics, PCA was performed (Figure 2), including all available data. Principal component 1 (PC 1) explained 61.8% of the variance and principal component 2 (PC 2) explained 21.6%, representing an 83.4% of all the variance. PC 1 was strongly correlated with leaf area (A), leaf perimeter (P), perimeter to area ratio of leaf ($P^2/A$), maximum length of leaf (LMax), minimum length of leaf (LMin), maximum length of petiole (LMax P) and red color, while PC 2 was strongly correlated only with red and green index (RG). Both components allowed discriminating the different grapevine varieties except Sauvignon Blanc, which was not related to any ampelographic parameter. Pinot Noir was positively related to eccentricity (Ex) and negatively related to LMax P. Carmenère was positively related to fractal dimension (FR) and negatively related to blue color. Merlot was positively related to most of the leaf morphometric parameters, such as A, LMax and LMin, whereas Chardonnay showed the opposite behavior, and it was negatively correlated to leaf morphometric characteristics.

In order to discriminate the different varieties and assess the effects of water stress conditions (leaf water potential $<-1.4$ MPa) on leaf a morpho-colorimetric characteristics, PCA was performed (Figure 3a). PC 1 (PC 1) explained 64.1% of the variance and principal component 2 (PC 2) explained 20.2%, representing 84.4% of all the variance. PC 1 was strongly correlated to A, P, $P^2/A$, LMax, LMin, LMax P and red color, while PC 2 was only strongly correlated with RG. Both components allowed discriminating the different grapevine varieties except for Sauvignon Blanc, which was not related to any morpho-colorimetric parameter. Under water stress conditions, Chardonnay and Pinot Noir were positively related to green color and negatively related to most of the leaf morphometric

parameters such as A, LMax, LMin and P$^2$/A, whereas Merlot showed the opposite behavior. Carmenère was negatively related to Ex and blue color.

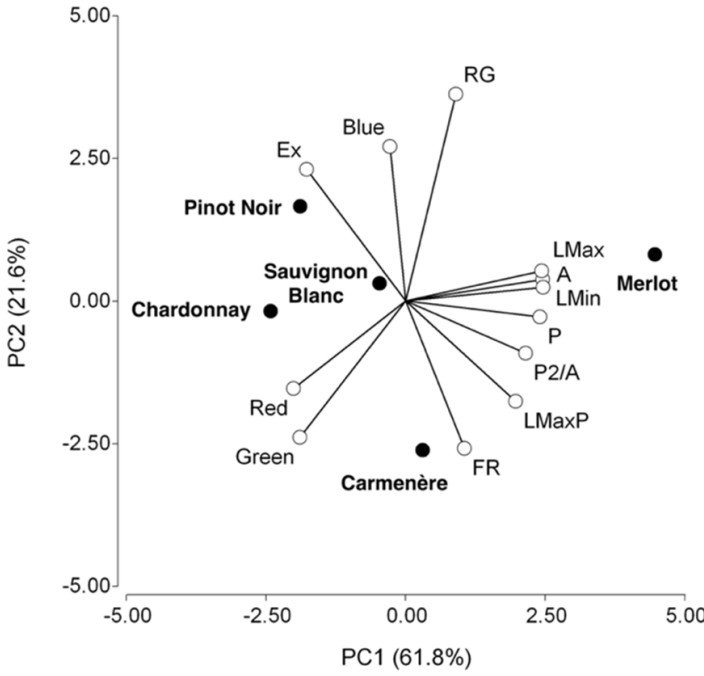

**Figure 2.** Principal component analysis (PCA) graph combining score plot and loadings for leaf morpho-colorimetric parameters obtained from five different grapevine varieties considering all data.

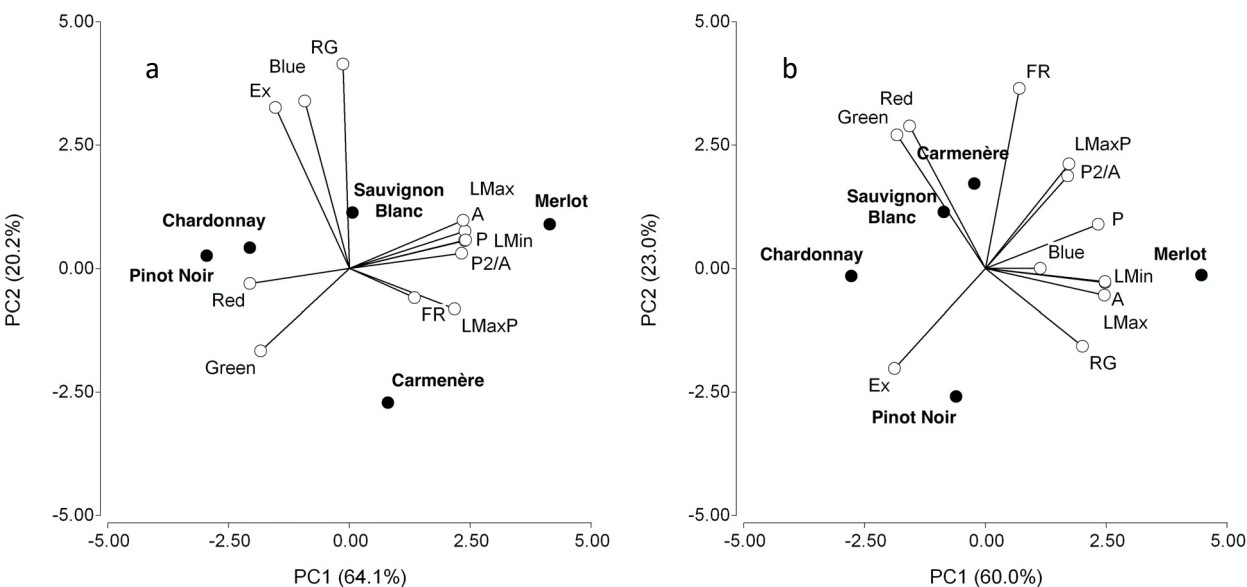

**Figure 3.** Principal component analysis (PCA) graph combining score plot and loadings for leaf morpho-colorimetric parameters obtained from five different grapevine varieties managed under stress water conditions (left, **a**) and no water restriction (right, **b**).

In order to discriminate the different varieties and assess the effects of no water stress (without irrigation restriction) on leaf morpho-colorimetric characteristics, PCA was performed (Figure 3b). Principal component 1 (PC 1) explained 60.0% of the variance, and principal component 2 (PC 2) explained 23.0%, representing 83.0% of all the variance. PC 1 was strongly correlated to A, P, LMax, LMin and red and green index, while PC 2 was only strongly correlated with FR. Both components allowed discriminating different

grapevine varieties except Carmenère, which was not related to any ampelographic parameter. Pinot Noir was positively related to Ex and negatively related to FR, whereas Sauvignon Blanc was negatively related to RG. Merlot was positively related to some leaf morphometric parameters such as A, LMax and LMin, whereas Chardonnay presented the opposite behavior.

## 4. Discussion

Since the beginning of the century, there is a new tendency within the wine industry to introduce varieties of different origins for wine diversification [19]. A correct ampelographic classification of grapevine varieties allows ensuring varietal authenticity and avoids planting material the wrong at a vineyard establishment. In the past, there have been varietal identification mistakes or confusion with grapevine material, both internationally and in the national wine industry. The Carmenère variety was improperly known as Merlot in Chile, and it was not until two decades ago that it was correctly identified by ampelography description [20]. Currently, Carmenère (7.87%) is the fifth most planted variety behind Cabernet Sauvignon (29.50%), Sauvignon Blanc (11.17%), Merlot (8.63%) and Chardonnay (8.16%) in Chile. Based on this, mistakes in grapevine identification potentially can result in significant financial costs associated with replants and in other aspects of the wine production chain.

Based on this, it is interesting that our results showed clear discrimination between Merlot and Carmenère using digital images and multivariate analysis (Figures 2 and 3). Merlot was strongly related to high values of leaf morphometric parameters, while Carmenère was inversely related to blue color and positively related to fractal dimension. In spring, the young leaves of Carmenère variety present a characteristic orange color, while the leaves take on an anthocyanin pigmentation that gives them a characteristic blue-reddish color when close to harvest [21]. The fractal dimension shows a strong correlation with human perception of surface roughness [22], which could discriminate Carmenère from the rest of the studied grapevine varieties after its analysis under both plant water status (Figure 2). On the other hand, Pinot Noir was positively related to eccentricity (Figure 2). The eccentricity is the ratio of the distance between the foci of the ellipse and its major axis length, and its value ranges between 0 and 1 [23]. If the grapevine varieties leaves are viewed from the eccentricity point of view, a high value of this parameter results in an elongated shape of the leaf. Pinot Noir leaf is of medium size, orbicular, dark green, thick, funnel-shaped, generally entirely trilobed or weakly trilobed [9]. These results are very interesting since some techniques do not readily discriminate varieties with morphological differences due to mutations, as in the case among Pinot Noir, Pinot Blanc and Pinot Gris [24].

Severe water stress (leaf water potential $<-1.4$ MPa) affected grapevine variety discrimination (Figure 3). Chardonnay and Pinot Noir were strongly positively related to red color and negatively related to leaf morphometric parameters, which is opposite to Merlot (Figure 3). This grapevine variety reached non-significant high values of net $CO_2$ assimilation compared to Chardonnay and Pinot Noir (Table 3). The red light is absorbed more strongly by photosynthetic pigments and is predominantly absorbed by the top few cell layers; consequently, red photons used less efficiency in terms of being dissipated [25]. Merlot presented non-significant higher intrinsic water-use efficiency (WUE) than the rest of the studied grapevine varieties. WUE reflects the balance between vine production (kg of biomass produced or moles of $CO_2$ assimilated) and water costs (m$^3$ of water used or moles of water transpired) [18,26]. Several reports showed that Merlot presents higher WUE compared to several grapevine varieties [13,26–28]. Therefore, the hydric behavior and gas exchange of Merlot vines grown under severe water stress could not affect their ability to develop a canopy with a high leaf area and, thus, is positively related to leaf morphometric parameters.

Non-water stress (without irrigation restriction) affected the discrimination of Sauvignon Blanc and Carmenère (Table 3). Sauvignon Blanc reached higher values of stomatal

conductance and transpiration and lower values of WUE compared to some of the rest of the studied varieties (Table 3). In Carmenère, transpiration could be sustained, despite presenting a large leaf area through less sensitive stomata [17]. In this case, the hydraulic system needs high-water transport capacity in the soil-plant-atmosphere continuum, and this variety can be less sensitive to xylem embolisms [29].

To solve the problems of discrimination of grapevine varieties by digital images, the selected leaf morpho-colorimetric parameters must not be affected by the environmental factors and viticultural practices. The color of upper size of blade, area of leaf anthocyanin coloration, goffering of blade, shape of teeth, size of teeth in relation to blade, length of teeth compared with their width, degree of opening and overlapping of petiole sinus and shape of base of petiole sinus, among other descriptors, could be interesting parameters for ampelographic classification of the grapevine varieties using digital images.

## 5. Conclusions

Leaf morpho-colorimetric variables studied in this trial allowed discriminating grapevine varieties, except Sauvignon Blanc, after their analysis under contrasting hydric conditions. Merlot, Carmenère, Chardonnay and Pinot Noir were separated by using digital images and multivariate analysis. Merlot was strongly related to high values of leaf morphometric parameters contrary to Chardonnay, while Carmenère was inversely related to blue color and positively related to fractal dimension. On the other hand, Carmenère and Pinot Noir were related to fractal dimension and eccentricity, respectively. RGB color system variables allowed discriminating the grapevine varieties managed under water stress conditions. Chardonnay and Pinot Noir were positively related to red color and negatively related to leaf morphometric parameters, which is opposite to Merlot. Full irrigation affected the discrimination of Sauvignon Blanc and Carmenère. Thus, leaf morpho-colorimetric characteristics of the vines were modified by plant water status. Lastly, the digital image is an interesting method to classify grapevine varieties, but it is important to validate it in future studies, proposing new leaf morpho-colorimetric variables to more clearly separate the varieties that cannot be discriminated, such as Sauvignon Blanc in this study.

**Supplementary Materials:** The following are available online at https://www.mdpi.com/article/10.3390/horticulturae7090315/s1, Table S1. Variable's contribution in the PCA combined (Figure 2). Table S2. Variable's contribution in the PCA grapevine varieties managed under stress water conditions (Figure 3a). Table S3. Variable's contribution in the PCA grapevine varieties managed under no water stress (Figure 3b).

**Author Contributions:** Conceptualization, C.A.-O. and H.V.-G.; methodology, N.T.-H. and M.A.-A.; software, N.T.-H. and M.A.-A.; validation, N.V.-V., M.A.-A., C.A.-O. and H.V.-G.; formal analysis, N.T.-H., G.G.-G., N.V.-V. and M.A.-A.; investigation, C.A.-O., H.V.-G. and Y.M.-S.; resources, C.A.-O., H.V.-G. and Y.M.-S.; data curation, N.T.-H., N.V.-V., C.A.-O., H.V.-G. and M.A.-A.; writing—original draft preparation, N.T.-H., G.G.-G., N.V.-V. and M.A.-A.; writing—review and editing, C.A.-O., H.V.-G. and Y.M.-S.; visualization, C.A.-O., H.V.-G. and Y.M.-S.; supervision, C.A.-O., H.V.-G. and Y.M.-S.; project administration, C.A.-O. and H.V.-G.; funding acquisition, C.A.-O. and H.V.-G. All authors have read and agreed to the published version of the manuscript.

**Funding:** This research received no external funding.

**Institutional Review Board Statement:** Not applicable.

**Informed Consent Statement:** Not applicable.

**Data Availability Statement:** Not applicable.

**Acknowledgments:** The authors would like to thank K.-H. Schulze, Maribel Rojas, Claudio Verdugo and all the technical staff of Panguilemo Experimental Station for their invaluable role in the good course of the experiments.

**Conflicts of Interest:** The authors declare no conflict of interest.

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
