# Peer review of "Leaf Morpho-Colorimetric Characterization of Different Grapevine Varieties through Changes on Plant Water Status"

_horticulturae, doi:10.3390/horticulturae7090315_

Round 1

Reviewer 1 Report

The paper presented a study to characterize different grapevine varieties according to their leaf morpho-colorimetric parameters. Digital images and multivariate analysis were used. The results showed that water stress could affect leaf morpho-colorimetry and fractal dimension parameters, and different grapevine varieties are related to various parameters. In general, the paper is easy to read and understand. The usefulness of the imaging and multivariate analysis was demonstrated in the study. 

A few aspects are not very clear in the paper:

- Were eight mature leaves collected per plant? If so, please clarify this in the paper. 

- The PCA results in Figures 1, 2, and 3 are not clear. 
(1) How was the position of each parameter (e.g., LMax, A, LMin, etc.) computed in each plot?
(2) How was the position of each variety (e.g., Pinot Noir, Merlot, etc.) computed in each plot?
(3) How to determine if a principal component was strongly correlated with a parameter? For example, from Figure 1, why was PC2 strongly correlated only with red and green index? And, why was PC1 strongly corrected with the parameter such as A, P, LMax, and LMin?
(4) The star coordinate type plot is used in Figure 1. What is its meaning?  

Other comments:
-It is desired that image results are included in the paper to help illustrate the related descriptions. For example, the image results can help readers understand "Based on this, if the grapevine varieties leaves are viewed from the eccentricity point of view, a high value of this parameter results in an elongated shape of the leaf." 

Author Response

The paper presented a study to characterize different grapevine varieties according to their leaf morpho-colorimetric parameters. Digital images and multivariate analysis were used. The results showed that water stress could affect leaf morpho-colorimetry and fractal dimension parameters, and different grapevine varieties are related to various parameters. In general, the paper is easy to read and understand. The usefulness of the imaging and multivariate analysis was demonstrated in the study.

Answer: Thank you very much for the comments, we value the good appreciation of the manuscript.

Material and Methods:

L117-173: Were eight mature leaves collected per plant? If so, please clarify this in the paper.

Answer: The text about eight mature leaves collected per plant has been included in lines 169-171.

L175-203: It is desired that image results are included in the paper to help illustrate the related descriptions. For example, the image results can help readers understand "Based on this, if the grapevine varieties leaves are viewed from the eccentricity point of view, a high value of this parameter results in an elongated shape of the leaf."

Answer: A sample of scanned leaves of the 5 cultivars for the classification study and corresponding images for automatic shape recognition to obtain morpho-colorimetric parameters from each leaf has been included in lines 183-185.

Results and Discussion: L303-324: The PCA results in Figures 2, 3, and 4 are not clear. How was the position of each parameter (e.g., LMax, A, LMin, etc.) computed in each plot?

Answer: The principal components analysis (PCA) generates the coordinates of the parameters (e.g. LMax, A, LMin, etc.) by means of the squared cosines of the variables.

 L303-324: How was the position of each variety (e.g., Pinot Noir, Merlot, etc.) computed in each plot?

Answer: The principal components analysis (PCA) generates the coordinates of the observations (in this case varieties Pinot noir, Sauvignon Blanc, etc.) by means of factorial scores.  

L293-303: How to determine if a principal component was strongly correlated with a parameter? For example, from Figure 1, why was PC2 strongly correlated only with red and green index? And, why was PC1 strongly corrected with the parameter such as A, P, LMax, and LMin?

Answer: Following the suggestions of Reviewer 1, the text “PCA was performed to visualize the classification of the leaves based on the parameters measured. The PCA was performed to identify the main relationships among the measured variables and to relate them variables to the quality of characterization (prediction) of the different grapevine varieties. Following this assumption, the proposed method considers that the first and second principal components of the PCA (PC 1 and PC 2) should represent the leaf morpho-colorimetric characterization at each observation (variety) if a significant amount of variance is explained by the PC 1 and PC 2. If so, the ccore of both components can be used to rank all the sampling sites according to leaf morpho-colorimetric parameters values observed at these sites.” has been added in lines 229-238.

 L293-303: The star coordinate type plot is used in Figure 1. What is its meaning? 

Answer: A biplot PCA was used where the vectors of each variable (average variance) were combined with each variety under study.

Reviewer 2 Report

The introduction section is short but in my opinion precise enough. I appreciate short and targeted introductions. The authors roughly define the state of the art and the problem they are seeking to cover but for it remains unclear how the work will help cover the gap thay have identified. The experimental design looks robust and correctly explained. I was not able to find the meaning of letters a, ab, etc. displayed in tables 1 and 2, for example. Do they refer to statistical significance of the Tukey´s test? Then it would be worth to define the statistical tests carried out. A more detailed description of the multivariate analysis conducted (PCA) is also missing. For me it would be very interesting to check how each variable contribute to each PCA, I mean the linear combination´s coefficients. Discussion and conclusions are correct.

Author Response

Introduction 

L217: The introduction section is short but, in my opinion, precise enough. I appreciate short and targeted introductions. The authors roughly define the state of the art and the problem they are seeking to cover but for it remains unclear how the work will help cover the gap they have identified.

Answer: We appreciate the valuable suggestions. Regarding the contribution made by the described vacuum work, it is related to validating an easy-to-implement methodology for different leaves of varieties subjected to different water regimes. The above is mentioned in the work objective in lines 85-88. 

Results L217: I was not able to find the meaning of letters a, ab, etc. displayed in tables 2 and 3, for example. Do they refer to statistical significance of the Tukey´s test? Then it would be worth to define the statistical tests carried out.

Answer: The text “For each variable and treatment, different letters in the same row show statistically significant differences assessed using the Tukey test (α = 0.05)” has been presented in lines 270-271 and 285-287. 

L289-337: A more detailed description of the multivariate analysis conducted (PCA) is also missing. For me it would be very interesting to check how each variable contribute to each PCA, I mean the linear combination´s coefficients. 

Answer: We agree with Reviewer 2, a more detailed description of the multivariate analysis conducted (PCA) was added in lines 229-238. Besides, the table for variable contributions was added as supplementary material and was placed in the text “Finally, the contribution of each variable to each principal component was added as supplementary material.” in lines 240-241.

Supplementary Table 1. Variable’s contribution in the PCA combined (Figure 2).

Parameter

PC 1

Correlation

PC 2

Area (A)

1.00

0.09

Perimeter (L)

0.98

-0.07

Maximum length (LMax)

0.99

0.13

Minimum length (LMin)

1.00

0.06

Maximum length of petiole (LMax P)

0.80

-0.42

Area to perimeter ratio (P2/A)

0.87

-0.22

Eccentricity (Ex)

-0.72

0.55

Red

-0.81

-0.37

Green

-0.77

-0.57

Blue

-0.11

0.65

RGB color scale

0.36

0.87

Fractal dimension (FR)

0.43

-0.62

Supplementary Table 2. Variable’s contribution in the PCA grapevine varieties managed under stress water conditions (Figure 3a).

Parameter

PC 1

Correlation

PC 2

Area (A)

0.98

0.18

Perimeter (L)

0.98

0.13

Maximum length (LMax)

0.97

0.23

Minimum length (LMin)

0.99

0.13

Maximum length of petiole (LMax P)

0.89

-0.19

Area to perimeter ratio (P2/A)

0.95

0.07

Eccentricity (Ex)

-0.63

0.75

Red

-0.85

-0.07

Green

-0.76

-0.38

Blue

-0.38

0.78

RGB color scale

-0.05

0.96

Fractal dimension (FR)

0.55

-0.14

Supplementary Table 3. Variable’s contribution in the PCA grapevine varieties managed under no water stress (Figure 3b).

Parameter

PC 1

Correlation

PC 2

Area (A)

0.99

-0.07

Perimeter (L)

0.94

0.22

Maximum length (LMax)

0.99

-0.13

Minimum length (LMin)

0.99

-0.07

Maximum length of petiole (LMax P)

0.69

0.53

Area to perimeter ratio (P2/A)

0.68

0.47

Eccentricity (Ex)

-0.76

-0.50

Red

-0.63

0.72

Green

-0.73

0.67

Blue

0.45

0.00

RGB color scale

0.81

-0.39

Fractal dimension (FR)

0.28

0.91

Round 2

Reviewer 1 Report

The revision has addressed my comments in the previous review round. Thank the authors for their effort to improve the paper.

This manuscript is a resubmission of an earlier submission. The following is a list of the peer review reports and author responses from that submission.